# Safety and Efficacy of Simultaneous Resection of Gastric Carcinoma and Synchronous Liver Metastasis—A Western Center Experience

**DOI:** 10.3390/medicina58121802

**Published:** 2022-12-07

**Authors:** Corina-Elena Minciuna, Stefan Tudor, Alexandru Micu, Andrei Diaconescu, Sorin Tiberiu Alexandrescu, Catalin Vasilescu

**Affiliations:** 1General Surgery Department, Fundeni Clinical Institute, 022328 Bucharest, Romania; 2Department of General Surgery, Faculty of Medicine, Carol Davila University of Medicine and Pharmacy, 050474 Bucharest, Romania

**Keywords:** gastric cancer, synchronous liver metastasis, liver resection, gastric resection

## Abstract

*Background and objectives*: Gastric cancer (GC) is often diagnosed in the metastatic stage. Palliative systemic therapy is still considered the gold standard, even for patients with resectable oligometastatic disease. The aim of the current study is to assess the potential benefit of up-front gastric and liver resection in patients with synchronous resectable liver-only metastases from GC (LMGC) in a Western population. *Materials and Methods*: All patients with GC and synchronous LMGC who underwent gastric resection with or without simultaneous resection of LMs between January 1997 and December 2016 were selected from the institutional records. Those with T4b primary tumors or with unresectable or more than three LMs were excluded from the analysis. All patients who underwent emergency surgery for hemorrhagic shock or gastric perforation were also excluded. *Results*: Out of 28 patients fulfilling the inclusion criteria, 16 underwent simultaneous gastric and liver resection (SR group), while 12 underwent palliative gastric resection (GR group). The median overall survival (OS) of the entire cohort was of 18.81 months, with 1-, 3- and 5-year OS rates of 71.4%, 17.9% and 14.3%, respectively. The 1-, 3- and 5-year OS rates in SR group (75%, 31.3% and 25%, respectively) were significantly higher than those achieved in GR group (66.7%, 0% and 0%, respectively; *p* = 0.004). Multivariate analysis of the entire cohort revealed that the only independent prognostic factor associated with better OS was liver resection (HR = 3.954, 95% CI: 1.542–10.139; *p* = 0.004). *Conclusions*: In a Western cohort, simultaneous resection of GC and LMGC significantly improved OS compared to patients who underwent palliative gastric resection.

## 1. Introduction

Gastric cancer (GC) is one of the major public health problems despite its decrease in incidence [1]. More than one third of patients with GC have metastases at the time of their primary tumor diagnosis [1]. The most common site for GC metastasis is the liver, with 41.3% of stage IV patients having liver involvement [2]. Most patients are treated with palliative oncologic therapy, although some recent evidence has suggested a survival benefit for patients who underwent complete resection of GC and metastatic disease sites [3,4,5,6]. Even in patients with oligometastatic disease (OMD), only a small percentage underwent curative intent surgery (0.4–2.3%) [1]. In the literature, there is not a universally accepted definition of OMD, although a consensus of ESRO and EORTC recommended taking into account the number of involved organs, the number and size of metastases, as well as the number of lesions per organ [7]. A recent meta-analysis regarding the definition of OMD in esophageal and gastric cancer revealed that among the published studies on this topic, more than 75% agreed (consensus) that OMD should include those patients with only single organ involvement. The number of metastases accepted for inclusion in the OMD group depended on the organ. Thus, in patients with liver oligometastasis, up to three liver (uni/bilobar) metastases should be present, in lung oligometastasis, up to two unilateral metastases should be present, while in extra-regional lymph node oligometastasis, one station should be involved [8]. For patients with peritoneal involvement, the definition of OMD is even less clear, with most studies recommending inclusion in this group of patients with a peritoneal cancer index up 6 (PCI ≤ 6) [9]. Taking into account these recommendations, in this study only patients with up to three liver-only metastases from GC were included.

According to the majority of Western guidelines, the gold-standard treatment for patients with GC and synchronous metastasis is palliative systemic therapy (immuno- and/or chemotherapy), even in those with oligometastatic disease [10,11]. Prior to chemotherapy, the median survival was around 5 months, being worse for those with liver and bone metastasis, whose median survival was 2 months [3,12]. Although palliative chemotherapy increased survival [12], the prognosis still remains poor. Even with modern therapies, such as immunotherapy, the median survival does not exceed 13.8 months [13]. These results encouraged some of the surgical centers, especially in Asia, to proceed with resection of primary tumor and synchronous metastases limited to an organ/site (e.g., liver, peritoneum) in stage IV GC patients [3,6,14,15]. These preliminary reports, mainly from uncontrolled case series, suggested a survival benefit after simultaneous resection of GC and synchronous liver-only metastases (LMGC). Therefore, recent guidelines from Asian countries/societies recommend the simultaneous resection (SR) of GC and synchronous LMGC in selected patients [16,17,18]. The Chinese consensus went further to better standardize the approach and proposed a new clinical classification system [19]. Thus, type I LMGC is limited to technically resectable 1–3 liver metastases, whose largest diameter does not exceed 4 cm, or LMGC located in one hemi-liver without involvement of major vessels or bile ducts [19]. The recommendation for this category is either up-front surgery or surgery after neoadjuvant systemic treatment. In type II LMGC (that comprised potentially resectable metastases out of the range of type I), the recommended therapeutic approach consists of initial systemic treatment followed by imaging reassessment and subsequent surgery for those without evidence of disease progression during oncologic therapy [19].

However, all these updates did not change the ESMO [11] and NCCN [10] recommendations for the treatment of patients with liver-limited metastases from GC. Due to the limited number of western series dealing with this subject, the potential benefit of liver resection for synchronous LMGC in a western population cannot be adequately estimated yet.

The aim of the current paper is to address this subject in a high-volume western center (for both liver resection and gastric resection). Although REGATTA [20] trial revealed that palliative gastrectomy did not improve overall survival (OS) in patients with synchronous limited metastases from gastric carcinoma, one may hypothesize that palliative gastrectomy may induce a bias in the overall survival of patients with synchronous LMGC. To avoid this bias in the evaluation of the benefit of simultaneous gastric and liver resection (SR), the current study compared the results of this approach to those achieved in similar patients treated by palliative gastrectomy without liver resection.

## 2. Materials and Methods

### 2.1. Data Source and Patient Selection

This is a retrospective observational study. All the patients with gastric carcinoma and synchronous liver metastases were selected from a prospectively maintained database, including all the patients operated on for gastric cancer (either with palliative or curative intent) in our center between January 1997 and December 2016. The patients with recurrent gastric carcinoma, primary invasive tumor in the adjacent organs (T4b stage) or concomitant extrahepatic metastases were excluded from the analysis. The patients whose gastric cancer was not resected and those operated on on an emergency basis due to gastric perforation or hemorrhagic shock were also excluded from the analysis. Furthermore, in an attempt to achieve a homogenous group, the CT scan data of these patients were reviewed and those with more than 3 liver metastases or deemed unresectable by two independent surgeons were excluded from the study group.

### 2.2. Surgical and Oncologic Management

The diagnosis of GC was established preoperatively by upper endoscopy in all patients and confirmed on the gastrectomy specimen. A contrast-enhanced CT scan of the thorax, abdomen and pelvis was performed for staging. Liver metastasis from GCs were considered synchronous if they were detected prior to or during the surgery addressed to the primary tumor [21].

For the patients operated on between 1997 and 2000, the treatment decision belonged to the surgeon. Since 2001, all the patients with preoperatively diagnosed LMGC had preoperative discussions in a multidisciplinary meeting.

In those patients diagnosed with LMGC preoperatively, surgery was recommended by a multidisciplinary team in order to manage the complications of the primary tumor (stenosis or hemorrhage/anemia). The decision to also perform liver resection belonged to the main surgeon. There were five surgeons who performed all the resections, each of them performing both gastrectomy alone and gastrectomy plus hepatectomy. All the surgeons were certified for both gastric surgery and hepatic surgery. The patients whose LMGC were not identified by preoperative CT scan underwent laparotomy with intention to perform a curative gastric resection. After intraoperative confirmation of LMGC, the decision to perform gastric resection (with or without liver resection) belonged to the main surgeon. The patients who underwent only palliative gastric resection (without liver resection) were included in the GR group. None of the patients who were treated by palliative gastrectomy further underwent liver resection. Those patients who underwent simultaneous gastric and liver resection were included in the SR group.

No patient in this cohort underwent neoadjuvant chemotherapy. All the patients underwent postoperative oncologic therapy. The postoperative chemotherapy consisted of an association of fluoropyrimidines with platinum-containing compounds, regardless of whether the liver metastases were resected or not. None of these patients was treated with Trastuzumab because the routine testing of HER2 receptor status in patients with GC started in our center in 2018. This was due to the fact that the National Health Insurance House has reimbursed the use of Trastuzumab in patients with gastric cancer and HER2-neu amplification since November 2017. Furthermore, in Romania, the use of anti-PD-1/PDL-1 therapy in the treatment of patients with metastatic GC is still not reimbursed by the National Health Insurance House. Thus, none of the patients included in this study received check-point inhibitors.

### 2.3. Ethics

Informed consent was obtained prior to surgery from all patients. The study protocol, number 21194/16.04.2021, was approved by the local Institutional Review Board.

### 2.4. Statistical Analysis

Categorical variables were reported as frequencies and percentages and were compared by Fisher’s Exact Test. Continuous variables were presented as median and interquartile range [IQR25–IQR75]. Overall survival (OS) represented the time interval between surgery and the death of the patient or the date of last follow-up if the patient was alive at that moment. The OS rates were estimated with the Kaplan–Meyer method, and they were compared between different groups by Log-rank test. Univariate analysis and multivariate analysis have been performed to assess the prognostic factors associated with OS, using Cox regression analysis. All variables associated with a *p* value ≤ 0.10 in the univariate analysis were included in the multivariate analysis, taking into consideration the size of the database. The differences with a *p*-value < 0.05 were considered statistically significant. All statistical analyses were performed using IBM SPSS Statistics for Windows (version 23.0, IBM, Chicago, IL, USA).

## 3. Results

### 3.1. Clinico-Pathological Characteristics

Twenty-eight patients met the inclusion criteria. In the entire cohort, most of the patients were males (75%) and the median age was 65.5 [54.5–71] years old. Regarding the T-stage, 3 patients (10.7%) had a T2 tumor, 21 (75%) had T3 gastric carcinoma and one patient (3.6%) had T4a GC (missing data in 3 patients). The primary tumor was located in the vertical part of the stomach in 17 (60.7%) patients. The degree of differentiation was G1 in 6 (21.4%) patients, G2 adenocarcinoma in 10 (35.7%), while 9 (32.1%) patients had G3 carcinomas (missing data in 3 patients). Considering lymph node involvement, six (21.4%) patients had N0 stage, four (14.3%) had N1 stage, seven (25%) had N2, five (17.9%) had N3a and two (7.1%) had N3b (missing data in four patients). Fifteen patients (53.6%) had one liver metastasis, 9 (32.1%) had two metastases and four (14.3%) had three liver metastases. The median diameter of the largest LMGC was 2 [1–3.9]. The maximum diameter of LMGC exceeded 3 cm in nine patients (four in the SR group and five in the GR group). In four patients, liver metastases were identified intraoperatively (three patients in the SR group and one patient in the GR group—*p* = 0.613, Fischer’s exact test); the diameter of these LMGC was smaller than 1 cm.

### 3.2. Surgical Data

Resection of the primary tumor consisted of total gastrectomy in 14 (50%) patients, distal gastrectomy in 11 (39.2%), proximal eso-gastrectomy in 2 (7.1%) cases, while one patient underwent a medio-gastric resection. D1 lymphadenectomy was performed in 22 (78.6%) patients, while D2 lymphadenectomy was associated with gastric resection in 6 (21.4%) patients (all of them receiving concomitant gastric and liver resection).

Out of these 28 patients, 16 underwent complete resection of malignant disease (simultaneous resection of primary tumor and liver metastases—SR group) and 12 patients underwent palliative gastric resection (GR group).

For the 16 patients who underwent liver resection associated with gastrectomy, the following types of hepatectomy were performed: non-anatomical wedge resections—14 patients (87.5%), left lateral sectionectomy—1 patient (6.25%) and right posterior sectionectomy—1 patient (6.25%).

Comparative clinico-pathological and surgical characteristics of the patients included in these two groups are presented in Table 1. There were no significant differences between the SR group and GR group, except for the type of lymphadenectomy (D2 lymphadenectomy was significantly more frequently performed in the SR group compared to the GR group—37.5% vs. 0%, respectively; *p* = 0.024).

### 3.3. Short-Term Outcomes

In the SR group, nine patients (56.3%) had no postoperative complications, five (31.3%) had grade I Dindo-Clavien complications, one (6.3%) developed grade II complication and one (6.3%) had grade IIIB Dindo-Clavien complication. The patient with the grade II complication developed postoperative pneumonia (receiving antibiotherapy) and the those with IIIB complication had hemorrhage from the liver resection site (undergoing reintervention for hemostasis). In the GR group, ten patients (83.3%) had no complications after surgery, while two patients developed Dindo-Clavien grade II complications (one had a surgical site infection requiring antibiotherapy and the other needed a postoperative blood transfusion for anemia). Although the percentage of patients who developed postoperative complications was higher in the SR group vs. the GR group (44% vs. 17%, respectively), the difference was not statistically significant (*p* = 0.115). There were no postoperative deaths, as described in Table 1.

### 3.4. Overall Survival (OS)

For the entire cohort, the median OS was 18.81 months, with 1-, 3- and 5-year OS rates of 67.85%, 17.85% and 14.28%, respectively.

For the SR group, the median survival was 25.83 months, significantly higher (*p* = 0.004) than for the GR group (15.13 months). The 1-, 3- and 5-year OS rates achieved by SR were 75%, 31.3% and 25%, respectively, while in the GR group the 1- and 3-year OS rates were 66.7% and 0%, respectively (Figure 1).

Because D2 lymphadenectomy was performed significantly more frequently in the SR group compared to the GR group (*p* = 0.024), the patients were also stratified according to the type of lymph node dissection (LND). Thus, in SR group, the patients who underwent D2 LND had a similar OS to those of patients with D1 LND (*p* = 0.423) (Figure 2). Furthermore, among patients with D1 LND, those in the SR group had a significantly higher median OS (24.53 months) compared to those in the GR group (15.13 months, *p* = 0.007) (Figure 3).

### 3.5. Univariate Analysis

For the entire cohort, in univariate analysis, the only factor that was significantly associated with higher OS was concomitant resection of the primary tumor and synchronous LMs (*p* = 0.002). The location of the primary tumor in the gastric antrum of the stomach was marginally associated (*p* = 0.10) with lower OS rates in univariate analysis (Table 2). Similarly, the maximum size of LMGC greater than 3 cm was marginally associated (*p* = 0.078) with lower OS (Table 2).

### 3.6. Multivariate Analysis

To identify the independent prognostic factors for better long-term outcomes, the variables that were associated with a *p* value ≤ 0.1 in univariate analysis were included in multivariate analysis. The only factor that was found to be independently associated with better OS was the simultaneous resection of GC and LMGC (HR = 3.954; 95% CI: 1.542–10.139, *p* = 0.004) (Table 2).

## 4. Discussion

This study suggests that simultaneous resection of gastric carcinoma and liver-only metastases was associated with significantly higher OS rates compared to the palliative resection of primary tumors. The control group in our study (GR group) included patients who underwent palliative gastric resection because few studies suggested that palliative gastrectomy associated with perioperative chemotherapy significantly improved OS compared to palliative oncologic therapy alone [22]. Although the survival benefit of palliative gastrectomy over palliative oncologic therapy has not been proven in randomized trials [20], it is undoubtable that the OS rates achieved in this control group could not be considered inferior to those of similar patients treated by palliative oncologic treatment alone [23]. A similar methodological approach was used in a recent meta-analysis which included 11 studies comparing the long-term outcomes of patients who underwent gastrectomy and hepatectomy vs. gastrectomy alone [24]. This meta-analysis, similar to other ones, found significantly higher OS rates achieved by gastrectomy and hepatectomy vs. gastrectomy alone in patients with either synchronous or metachronous LMGC [4,24]. Likewise, the results of our study support the survival benefit achieved by gastrectomy and liver resection in a western population with stage IV gastric cancer metastatic to the liver. This observation, along with the results of the few existing similar Western studies [5,13,21,25,26], could contribute to a paradigm-shift, taking into account the lack of recommendations for such an aggressive approach in most Western guidelines for the treatment of gastric cancer [10,11]. Furthermore, the 2018 Pan-Asian adapted ESMO Clinical Practice Guidelines for the management of patients with metastatic gastric cancer did not recommend the routine performance of metastasectomy in patients with stage IV GC [18]. Reflecting that paucity of recommendations for the resection of LMGC, a literature review found that hepatectomy was performed in only 0.4–1% of patients with LMGC [27]. Hence, there is a limited number of papers dealing with this subject, consisting of small retrospective studies and a few meta-analyses. Moreover, most of the studies concerning this issue came from Asian countries where this disease has a higher incidence [16,17]. Reports from Western countries are rare and have small samples size, this being an important problem given the different risk factors and gene pool. For example, out of 29 retrospective studies included in a recent meta-analysis which compared the impact on long-term outcomes of gastrectomy and liver resection to gastrectomy alone in patients with LMGC [4], only two studies came from Western centers. Those two studies included 21 (12 synchronous LMGC) and 31 (17 synchronous LMGC) patients, respectively [5,25]. In another meta-analysis [26], 9 out of 39 included studies reporting on Western cohorts of patients and only 3 of them had more than 30 patients. Furthermore, fewer than 5 Western studies published so far comprised more than 15 patients with resected synchronous LMGC each. Thus, our study has a sample-size comparable to those of most Western series and its results reflect the long-term outcomes achieved by liver resection in patients with synchronous LMGC in a Western center.

Although the current study included patients treated over a long period of time who did not receive the contemporary oncologic therapies, the 5-year OS rate of 25% is similar to those conveyed by most recent reports [4,28]. Furthermore, our encouraging long-term outcomes were achieved in patients with synchronous LMGC, who have significantly lower OS rates after liver resection, compared to patients with metachronous LMGC, according to most studies [4,13,21,29]. Therefore, in the meta-analysis of Granieri et al., patients undergoing liver resection for synchronous LMGC experienced a 5-year OS rate of 23.5%, significantly lower than those observed in patients with resected metachronous LMGC (29.2%; *p* = 0.001) [4].

Additionally, long-term outcomes achieved by simultaneous gastric and liver resection in the current study are higher than those reported by most Western series [5,21,25,30,31], except for the national series in England which found a 5-year OS of 38.5% [23]. The higher OS rates observed in our study may be explained, at least partially, by the selection criteria that were used. Thus, in the current study, only patients with T2–T4a primary tumors were enrolled, because in patients with GC invasive to adjacent organs, most authors considered that liver resection should be avoided due to the very poor OS rates achieved by surgery [30]. Furthermore, we included in our study only patients with 1–3 LMGC, and 69% of patients in the SR group had only one liver metastasis. This might explain the better OS rates achieved by liver resection in this study compared to the majority of Western reports, since most meta-analyses and retrospective studies revealed that patients with one or two resected LMGCs had significantly higher long-term outcomes than those with more liver metastases [4,5,28,32]. On the other hand, our long-term outcomes achieved by simultaneous resection of GC and LMGC in patients with three or fewer resectable LMGCs are in line with the recommendations of the Chinese consensus on the diagnosis and treatment of gastric cancer with liver metastases, which recommend, for type I LMGC, either upfront simultaneous resection or preoperative chemotherapy followed by simultaneous resection [19]. For patients with synchronous LMGC, preoperative chemotherapy is more frequently recommended by the European centers, whereas upfront surgery is the preferred approach in Japan, according to an international survey [33]. Furthermore, the real benefit of preoperative chemotherapy has not been clearly elucidated, due to the lack of comparative studies. The most important advantage of preoperative chemotherapy seems to be the better selection of patients [14], since responders had a higher disease-free survival rate than those with progressive disease [34].

Regarding the timing of gastric resection and liver resection, 89% of the centers (both from Europe and Japan) perform gastrectomy and hepatectomy during the same operation [33]. The current study also proved the feasibility and safety of the simultaneous resection of gastric tumors and liver metastases, as postoperative complications were not significantly higher in the SR group vs. the GR group. The rate of major complications of 6% and no postoperative mortality in patients undergoing simultaneous gastric and liver resection demonstrate that this approach could be safely performed in a high-volume center with highly specialized surgical and anesthesiology teams.

Of note, the results of current study should be regarded in light of its limitations. A major limitation of this study is its retrospective nature, which can induce an important bias. In an attempt to decrease the heterogenicity of this cohort, patients with T4b GC, as well as those with hemorrhagic shock or peritonitis due to gastric perforation, were excluded from this cohort. Furthermore, patients with unresectable LMGC or more than three resectable LMGCs or with extrahepatic metastases were also excluded from the current series. An indirect proof for achievement of a relative homogeneity of the two groups (SR group and GR group) is the fact that the only significant difference between them is the lower frequency of D2-lymphadenectomy in the GR group. However, no one study has proven that D2-lymphadenectomy improves survival in patients undergoing palliative gastrectomy so far. Moreover, in the present study, D2 lymphadenectomy has not been associated with better OS in the SR group. Another limitation of the current study is the lack of predetermined selection criteria for the resection of LMGC. Hence, the decision to proceed with hepatectomy always belonged to an experienced surgeon in both hepato-biliary and gastric surgery. The zero mortality observed in the current series is indirect proof of the expertise of surgeons, as previous studies reported an in-hospital mortality rate up to 2.6% in patients who underwent simultaneous resection of GC and LMGC [27,35]. The long period of the present study (including patients treated between 1997 to 2016) could also be perceived as a limitation, able to induce misleading results, especially due to changes in oncologic therapy during the last two decades [36]. However, in this study, the first-line postoperative chemotherapy consisted of the same regimen, regardless of whether the liver metastases were resected or not. Because the aim of the study was to compare the OS rates achieved by two types of surgical approaches, and the oncologic treatment offered to the patients treated in the same period was similar irrespective of the surgical approach, it may be considered that the changes in oncologic therapy over the study period had a limited impact on the comparison between the two groups, although a specific analysis on this topic has not been performed. Besides these, another limitation of the current study is the relatively small sample-size, both in the SR group and GR group. As we have already mentioned, most Western series which reported results achieved by simultaneous resection of GC and LMGC included less than 20 patients, with up to 5 studies reporting more than 15 patients. The lower number of patients also observed in the GR group reflects both the relatively strict inclusion criteria used in this study and the small number of patients who underwent palliative gastrectomy only. This is a consequence of the fact that many retrospective studies, as well as REGATTA randomized clinical trial [20], failed to prove a survival benefit for palliative gastrectomy alone in patients with OMD. Thus, little evidence supports the management of patients with GC and OMD with palliative gastrectomy, unless the complications of the primary tumor require its resection. To overcome these limitations, future retrospective multicentric studies, or, even better, randomized clinical trials, are needed to assess the real impact of such an aggressive surgical approach in these patients.

## 5. Conclusions

In a Western cohort of patients with gastric carcinoma and synchronous liver-limited resectable metastases, simultaneous resection of primary tumor and liver metastases, followed by chemotherapy significantly improved overall survival, compared to patients treated by palliative gastric resection and chemotherapy. This study also demonstrated the safety of simultaneous gastric and liver resection in a high-volume center. Although some clinico-pathologic features were correlated with long-term outcomes according to recent meta-analyses, the adequate selection of patients for such an aggressive approach should be elucidated by future trials including larger cohorts of patients. The benefit achieved by preoperative chemotherapy should be clarified in randomized controlled trials, such as FLOT 5 (NCT02578368).

## Figures and Tables

**Figure 1 medicina-58-01802-f001:**
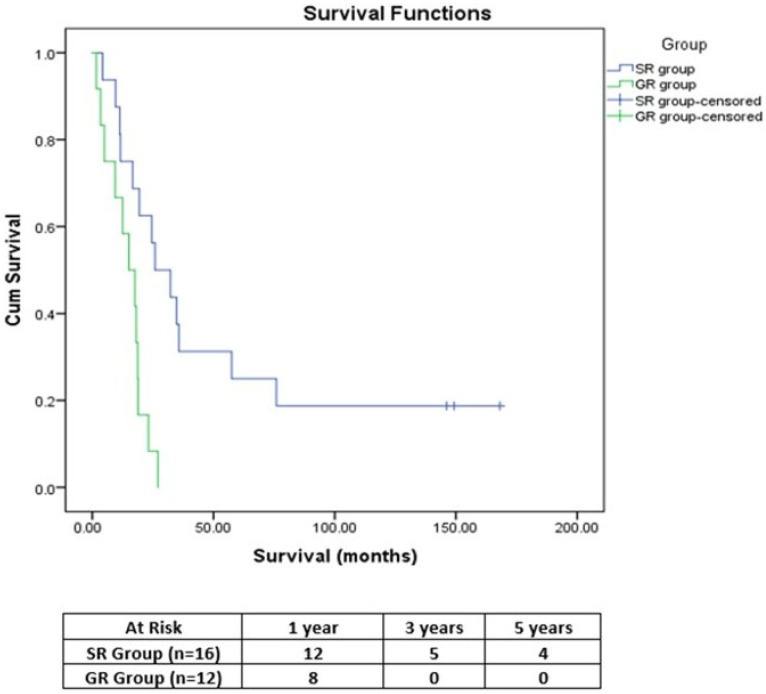
Comparative OS curves (SR group vs. GR group).

**Figure 2 medicina-58-01802-f002:**
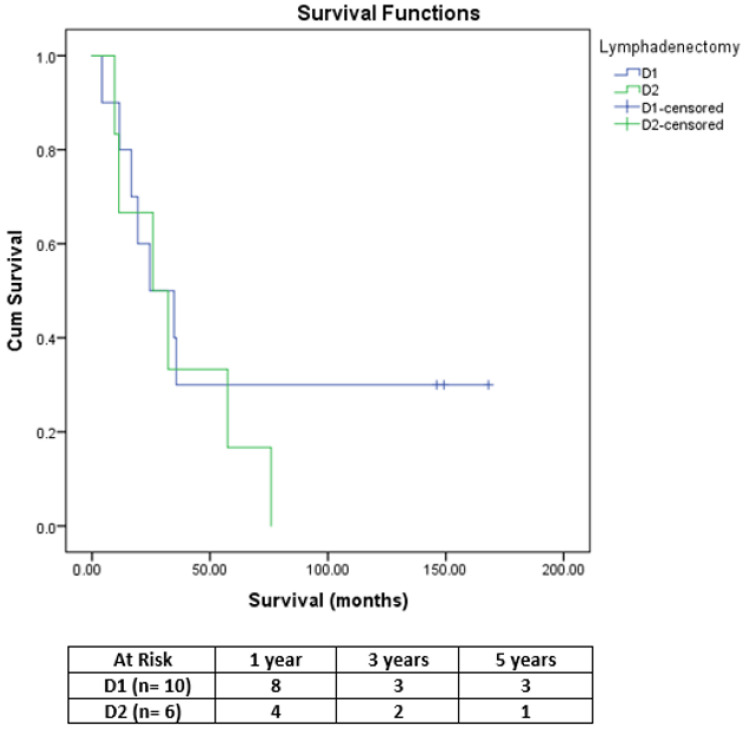
Comparative OS curves in SR group, according to the type of LND (D1 vs. D2).

**Figure 3 medicina-58-01802-f003:**
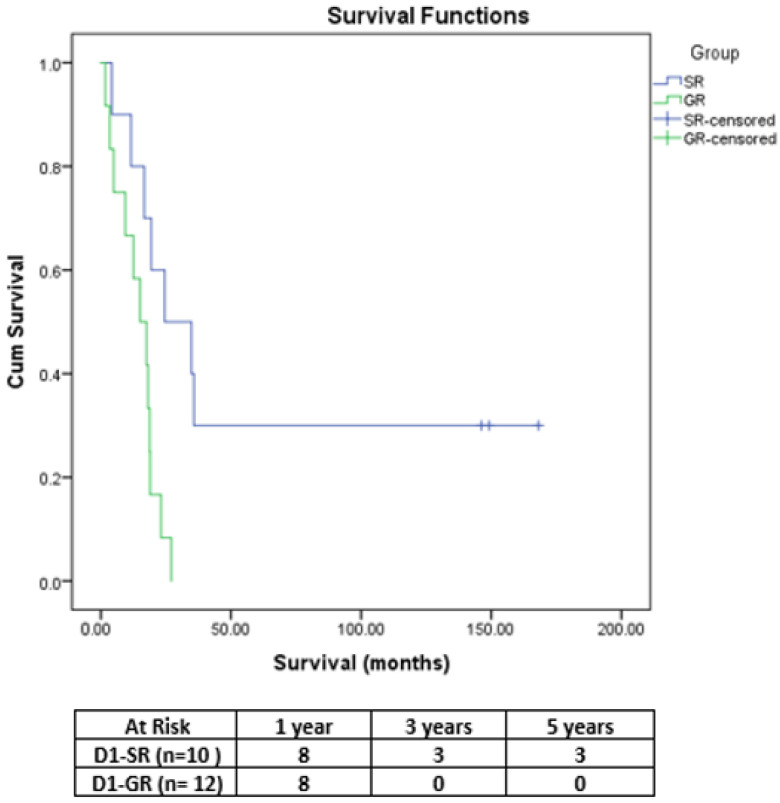
Comparative OS curves in patients with D1 LND (SR group vs. GR group).

**Table 1 medicina-58-01802-t001:** Comparative clinico-pathological characteristics between the SR group vs. GR group.

Comparative Descriptive Statistics				
		Group		
		SR	GR	*p* Value *
		Count (%)	Count (%)	
Gender	Female	5 (31%)	2 (17%)	0.661
	Male	11 (69%)	10 (83%)	
Age	<65 years	7 (43%)	5 (42%)	0.912
	≥65 years	9 (57%)	7 (58%)	
T	T2	2 (14%)	1 (9%)	0.491
	T3	12 (86%)	9 (82%)	
	T4a	-	1 (9%)	
	Missing	2	1	
N	N0	3 (21%)	3 (30%)	0.665
	N+	11 (79%)	7 (70%)	
	Missing	2	2	
Tumor site	Vertical part	12 (75%)	5 (42%)	0.121
	Gastric antrum	4 (25%)	7 (58%)	
Gastric Resection type	Total gastrectomy	10 (63%)	4 (33%)	-
	Partial gastrectomy	6 (37%)	8 (67%)	0.251
Number of liver metastases	1	11 (69%)	4 (33%)	0.143
	2	4 (25%)	5 (42%)	
	3	1 (6%)	3 (25%)	
Maximum diameter of liver metastases	≤3 cm	12 (75%)	7 (58.3%)	0.432
	>3 cm	4 (25%)	5 (41.7%)	
Lymphadenectomy type	D1	10 (63%)	12 (100%)	** *0.024* **
	D2	6 (37%)	0 (0%)	
G grade	1	4 (28%)	2 (18%)	0.809
	2	5 (36%)	5 (46%)	
	3	5 (36%)	4 (36%)	
	Missing	2	1	
Dindo-Clavien grade of postoperative complications	0	9 (56%)	10 (83%)	0.115
	I	5 (32%)	0	
	II	1 (6%)	2 (17%)	
	IIIb	1 (6%)	0	

* Fisher’s Exact Test was used to assess the differences between the groups.

**Table 2 medicina-58-01802-t002:** Univariate and multivariate analysis of risk factors associated with OS.

	Univariate Analysis			Multivariate Analysis		
	HR	95% CI	*p*-Value	HR	95% CI	*p*-Value
**T** T2 T3 T4a	1 1.93 2.94	0.446–8.351 0.251–34.379	0.379 0.390			
**N** N0 N+	1 0.839	0.323–2.184	0.719			
**Gender** Female Male	1 1.021	0.407–2.562	0.964			
**Age** <65 ≥65	1 1.395	0.622–3.130	0.419			
**Primary Tumor Site** Vertical Part Gastric antrum	1 2.070	0.868–4.939	0.101	1 1.360	0.378–4.898	0.638
**Maximum size of LMGC** ≤3 cm >3 cm	1 2.190	0.916–5.239	0.078	1 1.550	0.609–3.947	0.358
**Number of Liver Metastasis** 1 2 and 3	1 1.325	0.600–2.926	0.486			
**Lymphadenectomy** D1 D2	1 0.827	0.328–2.089	0.688			
**G grade** G1 G2 G3	1 2.184 0.801	0.735–6.488 0.253–2.539	0.160 0.706			
**Group SR vs. GR** SR GR	1 3.954	1.542–10.139	0.002	1 3.954	1.542–10.139	** *0.004* **
**Postoperative complications** 0 1	1 1.266	0.542–2.959	0.585			

## Data Availability

The datasets are not publicly available but are available from the corresponding author on reasonable request.

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
