# Peer review of "Safety and Efficacy of Simultaneous Resection of Gastric Carcinoma and Synchronous Liver Metastasis—A Western Center Experience"

_medicina, 2022, doi:10.3390/medicina58121802_

Round 1

Reviewer 1 Report

The manuscript is very interesting. The subject matter is currently being debated and may help define patients with gastric cancer and synchronous metastases as candidates for surgery.

Some points need to be clarified.

The first concerns the definition of oligometastatic disease: in the literature, this definition still needs to be clarified, and it must be specified that there is still a classification to be performed (both for liver metastases and for peritoneal carcinomatosis).

The second concerns the large amount of time patients are entered; although it is specified in the manuscript that the patients are distributed over a long period, it is important to highlight the new lines of chemotherapy for both epidermal growth factor receptor 2 (HER2) and programmed death-ligand 1 (PD-L1) used in the postoperative period as adjuvant therapy.

I suggest clarifying the definition of "horizontal part" (see line 203) to avoid confusion in less experienced readers: add, for example, "prepyloric or antral".

On line 282, you see the misspelling "in in Japan," double in in

I hope my suggestions are helpful.

Author Response

Dear Reviewer 1, thank you for your comments and recommendations which contributed to the improvement of the academic level of the manuscript.

Please find bellow the answers to your comments and recommendations. I also added the revised version of the manuscript.

  1. We added the next paragraph in Introduction and 3 new citations (7-9) on this topic.

“In the literature, there is not a universally accepted definition of OMD, although a consensus of ESRO and EORTC recommended to take into account the number of involved organs, the number and size of metastases, as well as the number of lesions per organ [7]. A recent meta-analysis regarding the definition of OMD in esophageal and gastric cancer revealed that among the published studies on this topic, more than 75% agreed (consensus) that OMD should include those patients with only one organ involvement. The number of metastases accepted for inclusion in OMD group depended on the organ. Thus, in patients with liver oligometastasis up to 3 liver (uni/bilobar) metastases should be present, in lung oligometastasis up to 2 unilateral metastases should be present, while in extra-regional lymph node oligometastasis one station should be involved [8]. For patients with peritoneal involvement, the definition of OMD is even less clear, most studies recommending inclusion in this group of patients with a peritoneal cancer index up 6 (PCI ≤ 6) [9]. Taking into account these recommendations, in this study were included only the patients with up to 3 liver-only metastases from GC.”

  1. We highlighted the actual use of Trastuzumab and check-point inhibitors in the actual treatment of patients with metastatic GC in the subchapter 2.2 “Surgical and oncological management” from “Materials and methods”. In this subchapter we added the following paragraph:

“None of these patients was treated with Trastuzumab, because the routine testing of HER2 receptor status in patients with GC started in our center in 2018. That was due to the fact that National Health Insurance House reimbursed the use of Trastuzumab in patients with gastric cancer and HER2-neu amplification since November 2017. Furthermore, in Romania, the use of anti-PD-1/PDL-1 therapy in the treatment of patients with metastatic GC is still not reimbursed by the National Health Insurance House. Thus, none of the patients included in this study received check-point inhibitors.”

  1. We changed the term “horizontal part” with “gastric antrum”, according to your recommendation.

  1. We made the correction on the line 282, by removing an “in” from the sentence “surgery is the preferred approach in Japan”.

Sincerely yours,

Sorin Alexandrescu, MD, PhD

General Surgery Department, Fundeni Clinical Institute, Bucharest, Romania

Carol Davila University of Medicine and Pharmacy, Bucharest, Romania

Reviewer 2 Report

I am attaching a review file

Author Response

Dear Reviewer 2, thank you for your comments and recommendations which contributed to the improvement of the academic level of the manuscript.

Please find bellow the answers to your comments and recommendations. I also added the revised version of the manuscript.

  1. We added the following paragraph in the subchapter “2.2. Surgical and oncologic management” of Materials and Methods section of the manuscript.

“For the patients operated between 1997 and 2000, the treatment decision belonged to the surgeon. Since 2001, all the patients with preoperatively-diagnosed LMGC were discussed preoperatively in a multidisciplinary meeting. In those patients diagnosed with LMGC preoperatively, surgery was recommended by multidisciplinary team in order to manage the complications of the primary tumor (stenosis or hemorrhage/anemia). The decision to perform also liver resection belonged to the main surgeon”.

  1. We added the following phrase in the subchapter “3.1. Clinico-pathological characteristics” of Results section of the manuscript:

“In 4 patients, liver metastases were identified intraoperatively (3 patients in SR-group and 1 patient in GR-group – p = 0.613, Fischer’s exact test); the diameter of these LMGC was smaller than 1 cm.”

  1. We added the following paragraph in the subchapter “2.2. Surgical and oncologic management” of Materials and Methods section of the manuscript.

“There were five surgeons who performed all the resections, each of them performing both gastrectomy alone and gastrectomy plus hepatectomy. All these surgeons were certified for both gastric surgery and hepatic surgery”.

  1. We added the following paragraph in the subchapter “2.2. Surgical and oncologic management” of Materials and Methods section of the manuscript.

“The postoperative chemotherapy consisted in association of fluoropyrimidines with platinum-containing compounds, regardless of whether the liver metastases were resected or not. None of these patients was treated with Trastuzumab, because the routine testing of HER2 receptor status in patients with GC started in our center in 2018. That was due to the fact that National Health Insurance House reimbursed the use of Trastuzumab in patients with gastric cancer and HER2-neu amplification since November 2017. Furthermore, in Romania, the use of anti-PD-1/PDL-1 therapy in the treatment of patients with metastatic GC is still not reimbursed by the National Health Insurance House. Thus, none of the patients included in this study received check-point inhibitors.”

  1. We added the following sentence in the subchapter “2.2. Surgical and oncologic management” of Materials and Methods section of the manuscript.

“None of the patients who were treated by palliative gastrectomy further underwent liver resection”.

  1. Because the extent of lymph node dissection (LND) was significantly different between the SR-group and GR-group, we stratified the patients, according to the type of LND. Thus, in the SR-group, the OS of patients who underwent D1 LND was similar to those of patients treated by D2 LND (p = 0.423). When compared the OS of patients who underwent D1 LND, those in the SR-group had a significantly higher median survival than those in GR-group. Altogether, that two observations revealed that D2 LND was not a significant factor correlated with OS. We reported these findings in the subchapter “3.4. Overall survival” from the Results section of the manuscript. Furthermore, we added two figures (Figure 2 and Figure 3) which depict the comparative OS rates according to the extent of LND.

“Because D2 lymphadenectomy was performed significantly more frequent in SR group compared to GR group (p = 0.024), the patients were also stratified according to the type of lymph node dissection (LND). Thus, in SR group, the patients who underwent D2 LND had a similar OS to those of patients with D1 LND (p = 0.423) (Figure 2). Furthermore, among patients with D1 LND, those in the SR group had a significantly higher median OS (24.53 months) compared to those in the GR group (15.13 months, p = 0.007) (Figure 3).”

Figure 2. Comparative OS curves in SR group, according to the type of LND (D1 vs. D2)

Figure 3. Comparative OS curves in patients with D1 LND (SR-group vs. GR-group) 

  1. We added the following paragraph in the subchapter “2.4. Statistical analysis” of Materials and Methods section of the manuscript.

“Categorical variables were reported as frequencies and percentages and were compared by Fisher’s Exact Test. Continuous variables were presented as median and interquartile range [IQR25-IQR75]”.

  1. We added the following paragraph in the subchapter “3.1. Clinico-pathological characteristics” of Results section of the manuscript:

“The median diameter of largest LMGC was 2 [1-3.9]. The maximum diameter of LMGC exceeded 3 cm in 9 patients (4 in SR-group and 5 in GR-group).”.

Furthermore, we introduced in the Table 1 a comparison of the largest diameter of LMGC between SR-group and GR-group. In SR-group were 4 patients with LMGC larger than 3 cm and 12 patients with LMGC up to 3 cm.  In GR-group were 5 patients with LMGC larger than 3 cm and 7 patients with LMGC up to 3 cm. The difference was not statistically significant: p = 0.432 (Fischer’s exact test) (see Table 1).

  1. We performed the stratification according to the extent of LND – see the answer to the question number 6.

  1. We added in the univariate analysis the maximum size of liver metastases (≤3 cm vs. > 3 cm) (Table 2). The maximum diameter of LMGC > 3 cm was marginally significant associated with lower OS in univariate analysis (p = 0.078). We added the following paragraph to the subchapter “3.5 Univariate analysis” of the Results section:

“Similarly, the maximum size of LMGC higher than 3 cm was marginally associated (p = 0.078) with lower OS (Table 2).”

However, in multivariate analysis, this factor (largest diameter of liver metastases) was not independently associated with OS (HR = 1.550; 95% CI: 0.609-3.947, p = 0.358) (Table 2).

Regarding the oncomarkers, the preoperative level of CEA or CA 19-9 was not available in most patients, thus we cannot evaluate their impact on OS. Regarding the postoperative chemotherapy, all the patients received as the first-line treatment a combination of fluoropyrimidines with platinum-containing compounds. Furthermore, none of the patients included in this study received Trastuzumab or check-point inhibitors (see the answer to the question nr. 4).

  1. We reformulated the assumption “This study demonstrated ….” as: “This study suggests that simultaneous resection of gastric carcinoma and liver-only metastases was associated with significantly higher OS rates compared to the palliative resection of primary tumor”.

  1. The phrase “Even the Pan-Asian adapted ESMO Clinical Practice Guidelines for the management of patients with metastatic gastric cancer consider that gastrectomy in patients with stage IV disease or metastasectomy are not routinely recommended” was reformulated as: “Furthermore, the 2018 Pan-Asian adapted ESMO Clinical Practice Guidelines for the management of patients with metastatic gastric cancer did not recommend the routinely performance of metastasectomy in patients with stage IV GC”.

  1. We made the change from “….. should be avoid …..” to “… should be avoided …”.

  1. Because we introduced new data about the oncologic treatment received by the patients included in this study, we reformulated the following phrase from the limitations part of Discussions:

“Because the aim of the study was to compare the OS rates achieved by two types of surgical approaches, and the oncologic treatment offered to the patients treated in the same period was similar irrespective the surgical approach, it may be considered that the changes in oncologic therapy over the study period had limited impact on comparison between the two groups, although a specific analysis on this topic have not been performed”.

Sincerely yours,

Sorin Alexandrescu, MD, PhD

General Surgery Department, Fundeni Clinical Institute, Bucharest, Romania

Carol Davila University of Medicine and Pharmacy, Bucharest, Romania

Reviewer 3 Report

This article comparing the short and long term outcomes of patients with gastric cancer and liver metastases after gastrectomy with or without hepatectomy is very interesting. Well written and well structured. The only limitation of this study is the number of patients

Author Response

Dear Reviewer 3, thank you for your comments!

Please find bellow the answers to your comments. I also added the revised version of the manuscript.

I fully agree with you, but most western studies dealing with this subject included a very limited number of patients. For sure, future multi-institutional studies are needed, trying to overcome this limitation.  We discussed this limitation in the Discussion chapter (highlighted in yellow).

“Reports from the Western countries are rare and have small samples size. For example, out of 29 retrospective studies included in a recent meta-analysis which compared the impact on long-term outcomes of gastrectomy and liver resection to gastrectomy alone in patients with LMGC[4], only two studies came from Western centers. Those two studies included 21 (12 synchronous LMGC) and 31 (17 synchronous LMGC) patients, respectively[5,22]. In another meta-analysis[23], 9 out 39 included studies reported Western cohorts of patients and only 3 of them had more than 30 patients. Furthermore, less than 5 Western studies published so far comprised more than 15 patients with resected synchronous LMGC, each. Thus, our study has a sample-size comparable to those of most western series.

Although some clinico-pathologic features were correlated with long-term outcomes according to recent meta-analyses, the adequate selection of patients for such an aggressive approach should be elucidated by future trials, including larger cohorts of patients.”

Sincerely yours,

Sorin Alexandrescu, MD, PhD

General Surgery Department, Fundeni Clinical Institute, Bucharest, Romania

Carol Davila University of Medicine and Pharmacy, Bucharest, Romania

Round 2

Reviewer 2 Report

I really appreciated the improvement the authors did at their paper. I think as it is, the work really deserves to be published, giving an important insight to a concept still not largely discussed in literature.

Author Response

Dear Reviewer

Thank you very much for your comments and recommendations (made in round 1), which helped us to improve the quality of the manuscript!

Sincerely yours,

Sorin Alexandrescu